# SMoP: Towards Efficient and Effective Prompt Tuning
## with Sparse Mixture-of-Prompts

**Joon-Young Choi[1], Junho Kim[1], Jun-Hyung Park[2], Wing-Lam Mok[1], SangKeun Lee[1,3]**

[1]Department of Artificial Intelligence, Korea University, Seoul, Republic of Korea
[2]BK21 FOUR R&E Center for Artificial Intelligence, Korea University, Seoul, Republic of Korea
[3]Department of Computer Science and Engineering, Korea University, Seoul, Republic of Korea
{johnjames, monocrat, irish07, wlmokac, yalphy}@korea.ac.kr

## Abstract

Prompt tuning has emerged as a successful parameter-efficient alternative to the full fine-tuning of language models. However, prior works on prompt tuning often utilize long soft prompts of up to 100 tokens to improve performance, overlooking the inefficiency associated with extended inputs. In this paper, we propose a novel prompt tuning method **SMoP** (**S**parse **M**ixture-**o**f-**P**rompts) that utilizes short soft prompts for efficient training and inference while maintaining performance gains typically induced from longer soft prompts. To achieve this, **SMoP** employs a gating mechanism to train multiple short soft prompts specialized in handling different subsets of the data, providing an alternative to relying on a single long soft prompt to cover the entire data. Experimental results demonstrate that **SMoP** outperforms baseline methods while reducing training and inference costs. We release our code at https://github.com/jyjohnchoi/SMoP.

## 1 Introduction

Prompt tuning (Lester et al., 2021; Liu et al., 2021) has recently gained attention as a parameter-efficient alternative to the full fine-tuning of language models. By freezing the original language model parameters and solely tuning the soft prompts (i.e., learnable token embeddings) added to the model input, prompt tuning achieves comparable performance to full fine-tuning while largely reducing the number of trainable parameters. Moreover, prompt tuning stands out for its conceptual simplicity and flexibility among other parameter-efficient fine-tuning methods (Houlsby et al., 2019; Guo et al., 2021; Hu et al., 2022), as it does not require modifications to the model structure.

Since the proposal of prompt tuning, there has been active research to enhance its efficiency and effectiveness. On one hand, several approaches propose to improve the performance of prompt tuning by integrating soft prompts into activations in each

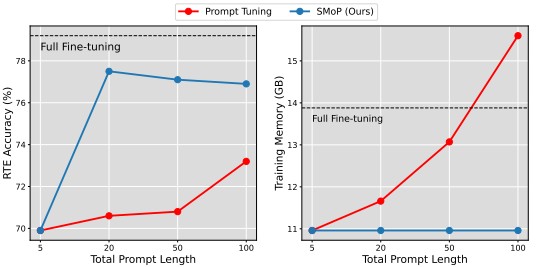

Figure 1: Accuracy (left) and training memory usage (right) with varying total prompt length on RTE dataset. For prompt tuning (Lester et al., 2021), increasing soft prompt length improves accuracy, but also results in a significant increase in memory usage. **SMoP** outperforms prompt tuning while preserving memory usage by sparsely activating short (length 5) prompts.

layer of the model (Li and Liang, 2021; Qin and Eisner, 2021; Liu et al., 2022), incorporating input-specific soft prompts (Jiang et al., 2022; Wu et al., 2022), or pruning and rewinding soft prompts (Ma et al., 2022). On the other hand, methods such as FPT (Huang et al., 2022) demonstrate improved training efficiency of prompt tuning in terms of convergence speed via progressive training.

Although these methods have empirically shown improvements in prompt tuning, they have overlooked the inefficiency associated with the extension of input sequences caused by the inclusion of soft prompts. While increasing soft prompt length (typically up to 100 tokens) is known to benefit model performance (Lester et al., 2021; Jiang et al., 2022), it consequently yields longer input sequences, leading to increased computational requirements during training and inference (see Figure 1). Therefore, we aim to investigate the utilization of relatively short soft prompts while preserving performance gains typically achieved from longer soft prompts.

To this end, we propose **SMoP** (**S**parse **M**ixture-**o**f-**P**rompts), a novel prompt tuning method that utilizes short soft prompts during training and in-

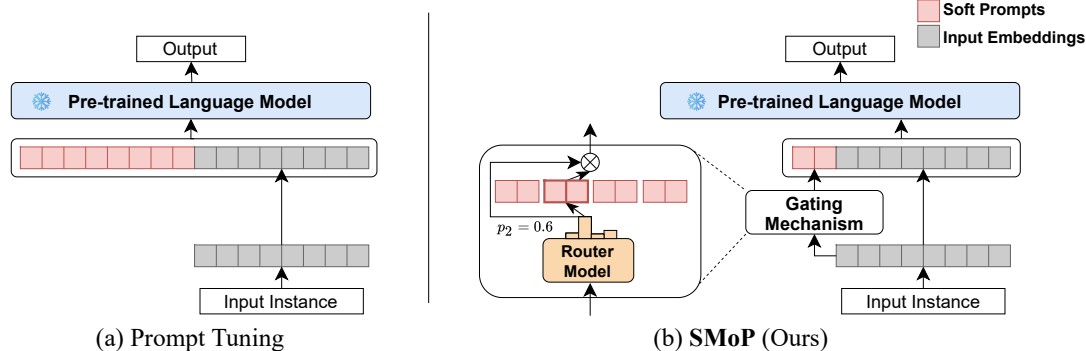

Figure 2: (a) Illustration of prompt tuning (Lester et al., 2021). A soft prompt is concatenated with the embedding representations of an input instance, and the soft prompt is solely fine-tuned. Given a soft prompt of 100 tokens, the length of the soft prompt is typically longer or similar to the input instance. (b) Illustration of our proposed method **SMoP**. A gating mechanism is employed to route each input instance to a short soft prompt.

ference. Given that using a single short soft prompt leads to inferior performance compared to longer soft prompts, our key insight is to train multiple short soft prompts that are specialized in handling different subsets of the data. To achieve this, we draw inspiration from the Sparsely-Gated Mixture-of-Experts (Shazeer et al., 2017; Fedus et al., 2022) that sparsely activates sub-networks (i.e., experts) to increase model capacity without a proportional increase in computation. We integrate this concept in the context of prompt tuning by employing a gating mechanism in **SMoP**, which guides each input instance to one of the short soft prompts based on its embedding representation. Such sparse activation enables effective utilization of short soft prompts without a significant increase in computation or degradation in performance.

To verify the efficiency and effectiveness **SMoP** introduces to prompt tuning, we conduct evaluations on six natural language understanding tasks from the SuperGLUE benchmark. Experimental results demonstrate that **SMoP** outperforms prompt tuning with reduced training and inference costs. In particular, **SMoP** improves the average performance of prompt tuning on six SuperGLUE tasks by 2.5%p with T5-base, and 3.4%p with T5-large on average while reducing training time, memory, and inference computations.

Our contributions are as follows:

1. We propose a novel prompt tuning method **SMoP** (**S**parse **M**ixture-**o**f-**P**rompts) that utilizes short soft prompts for efficient training and inference while maintaining performance gains often induced by increased soft prompt length.

2. **SMoP** sparsely activates short soft prompts via a gating mechanism that routes each instance to one of the multiple soft prompts based on its embedding representation.

3. Experimental results demonstrate that **SMoP** outperforms the baselines on T5-base and T5-large while utilizing shorter soft prompts, thereby using less training and inference costs.

## 2 Method

### 2.1 Preliminaries

**Full Fine-tuning** Assume that we have a sequence-to-sequence model $p_\phi(y \mid x)$ parameterized by $\phi$. Given an instance with a length $n$ sequence of embedding representations $X = \{x_1, x_2, ..., x_n\} \in \mathbb{R}^{n \times e}$ and corresponding label token embedding sequence $Y$, the objective function for full fine-tuning the model $p_\phi$ is as follows:

$$\arg\max_{\phi} \log p_\phi(Y \mid X). \tag{1}$$

**Prompt Tuning** If we define a length $l$ soft prompt with embedding dimension $e$ as $P_\theta$ which is parameterized by $\theta \in \mathbb{R}^{l \times e}$, the objective function of prompt tuning is as follows:

$$\arg\max_{\theta} \log p_\phi(Y \mid [P_\theta; X]), \tag{2}$$

where ; indicates the concatenation of the two matrices. Note that the language model parameters $\phi$ are no longer updated. Figure 2 (a) depicts the process of prompt tuning.

| Model | Method | Total Prompt Length | Utilized Prompt Length | Trainable Params (%) | Training Costs (↓) | | Inference Costs (↓) | Average Score (%) |
|---|---|---|---|---|---|---|---|---|
| | | | | | Time (s/100 steps) | Memory (GB) | FLOPs (GFLOPs/sample) | |
| T5-base | Full Fine-tuning | - | - | 100 | 81.4 | 14.9 | 70.3 | $78.2_{1.3}$ |
| | Prompt Tuning | 100 | 100 | 0.0344 | 70.7 (-13.1%) | 14.4 (-3.4%) | 98.1 (+39.5%) | $73.3_{1.9}$ |
| | P-tuning | 20 | 20 | 0.1028 | 64.9 (-20.3%) | 12.3 (-17.4%) | 76.6 (+9.0%) | $75.2_{2.1}$ |
| | **SMoP** (Ours) | 20 | 5 | **0.0083** | **61.0** (-25.1%) | **11.9** (-20.1%) | **71.5** (+1.7%) | $\mathbf{75.8}_{1.9}$ |
| T5-large | Full Fine-tuning | - | - | 100 | 176.1 | 29.2 | 247.8 | $83.4_{1.3}$ |
| | Prompt Tuning | 100 | 100 | 0.0139 | 151.2 (-14.1%) | 29.3 (+0.3%) | 378.1 (+52.6%) | $78.6_{1.4}$ |
| | P-tuning | 20 | 20 | 0.0407 | 131.9 (-25.1%) | 23.6 (-19.2%) | 291.6 (+17.7%) | $81.3_{1.8}$ |
| | **SMoP** (Ours) | 20 | 5 | **0.0033** | **129.1** (-26.7%) | **22.6** (-22.6%) | **275.4** (+11.1%) | $\mathbf{82.0}_{1.3}$ |

Table 1: Experimental results on six SuperGLUE tasks. Average training costs, inference costs, and performance for baselines and **SMoP** are presented. The percentage value next to each cost value indicates relative changes in cost values compared to full fine-tuning, and the subscript of the average score indicates the corresponding standard deviation. The highest performance and lowest cost values among prompt tuning methods are **bold** highlighted.

## 2.2 SMoP: Sparse Mixture-of-Prompts

The goal of **SMoP** is to train multiple short soft prompts, where each prompt is specialized in a subset of the data. To achieve this, **SMoP** employs a gating mechanism to direct the input instance to one of the soft prompts based on its embedding representations, as shown in Figure 2 (b).

In the gating mechanism, we introduce a small linear router model $L_\mu$ parameterized by $\mu \in \mathbb{R}^{e \times k}$ which makes decisions regarding which of the soft prompts the input should be routed to. Formally, given $k$ soft prompt embeddings $P_{\theta_1}, P_{\theta_2}, ..., P_{\theta_k}$ which are parameterized by $\{\theta_j\}_{j=1}^k$ where $\theta_j \in \mathbb{R}^{l \times e}$, the router model takes the average of input embeddings $\bar{X} \in \mathbb{R}^e$ as its input and calculates the routing probability $p_1, p_2, ..., p_k$ for each soft prompt. Thus, the routing probability of the $j$-th prompt is calculated as:

$$p_j(X) = [\text{softmax}(L_\mu(\bar{X}))]_j. \quad (3)$$

The input is then routed to the soft prompt with the highest probability, and the final soft prompt to be utilized is obtained as the product of the routed prompt and the probability value. Therefore, the objective function of **SMoP** is defined as follows:

$$\arg\max_{\mu, \theta_c} \log p(Y \mid [\, p_c(X) \cdot P_{\theta_c}; X]), \quad (4)$$

where c is the index of the prompt with the highest probability value. Note that in **SMoP**, while the total prompt length is $k \cdot l$, the utilized prompt length remains as $l$.

## 2.3 Router Perturbation

Prior works on Mixture-of-Experts (Chen et al., 2022b; Fedus et al., 2022) demonstrate that load balance among experts during training plays an important role in performance. To ensure load balance among soft prompts by encouraging exploration of inputs over diverse prompts, we apply router perturbation during the training of **SMoP**. Specifically, we add a scaled Gaussian noise $\delta \sim \mathcal{N}(0, 1)$ to the output value of the router model during training. Therefore, equation (3) is modified as follows:

$$p_j(X) = [\text{softmax}(L_\mu(\bar{X}) \circ (\vec{1} + \delta))]_j. \quad (5)$$

## 3 Experiments

### 3.1 Experimental Settings

**Tasks** To cover diverse NLP tasks in our experiments, we evaluate **SMoP** and baseline methods on six tasks[1] from the SuperGLUE benchmark (Wang et al., 2019). As the official test sets for Super-GLUE benchmark are not publicly released, we follow Chen et al. (2022a) to use the validation set as the test set and split the original train set into train and validation sets by 90%/10% proportion.

**Models and Baselines** Our experiments are built on the public HuggingFace (Wolf et al., 2019) implementation and pre-trained checkpoints of T5 (Raffel et al., 2020) in two scales: base and large.

To demonstrate the advantages that **SMoP** introduces to prompt tuning, we compare **SMoP** to prompt tuning (Lester et al., 2021), P-tuning (Liu et al., 2021), and full fine-tuning.

**Evaluation Setup** For prompt tuning methods, we experiment on {5, 20, 50, 100} soft prompt tokens, and for **SMoP**, we sweep through {2, 4, 10, 20} prompts of length {1, 3, 5, 10}. We report experimental results on the setting with the best average performance over two or three runs, as

---

[1]BoolQ, CB, COPA, MultiRC, RTE, WiC

| Model | $k$ / $l$ | 2 | 4 | 10 | 20 |
|---|---|---|---|---|---|
| T5-base | 1 | $72.9_{2.1}$ | $74.0_{1.5}$ | $73.5_{1.3}$ | $73.9_{1.2}$ |
| | 3 | $74.2_{2.3}$ | $74.0_{2.4}$ | $74.8_{2.4}$ | $74.2_{1.7}$ |
| | 5 | $75.0_{2.0}$ | $\mathbf{75.8}_{1.9}$ | $75.3_{1.8}$ | $74.7_{1.7}$ |
| | 10 | $75.1_{1.7}$ | $74.8_{1.7}$ | $75.2_{1.4}$ | $74.1_{2.0}$ |

Table 2: Average performance (%) on six tasks from the SuperGLUE benchmark with diverse utilized prompt lengths ($l$) and the number of prompts ($k$).

| Model | Method | BoolQ | CB | RTE | Average |
|---|---|---|---|---|---|
| T5-base | **SMoP (Ours)** | $79.4_{0.3}$ | $\mathbf{94.6}_{1.8}$ | $\mathbf{77.5}_{3.2}$ | $\mathbf{83.8}_{2.1}$ |
| | w/o perturbation | $\mathbf{79.7}_{0.2}$ | $93.5_{2.7}$ | $76.0_{1.5}$ | $83.1_{1.8}$ |
| | Top-2 | $78.4_{0.2}$ | $88.1_{1.0}$ | $69.7_{0.4}$ | $78.7_{0.6}$ |
| | Gumbel-Softmax | $79.2_{0.4}$ | $92.3_{2.0}$ | $75.2_{4.3}$ | $82.2_{2.7}$ |
| | Stochastic | $78.2_{0.3}$ | $86.9_{2.1}$ | $69.2_{1.7}$ | $78.1_{1.6}$ |
| | Single | $78.5_{0.0}$ | $89.3_{1.8}$ | $69.9_{0.8}$ | $79.2_{1.1}$ |

Table 3: Experimental results (%) on diverse routing methods for **SMoP**.

well as the corresponding standard deviations. We report training time[2] and memory usage as training costs and inference FLOPs as inference costs.

## 3.2 Results

### 3.2.1 Main Results

Table 1 presents the performance of **SMoP** and the baseline methods. Notably, **SMoP** achieves the highest performance among the baseline prompt tuning methods on SuperGLUE tasks with the least training and inference costs. On T5-base, **SMoP** demonstrates an average improvement of 2.5%p, while on T5-large, the improvement reaches 3.4%p. The detailed results of SuperGLUE tasks are shown in Appendix D.

The fact that **SMoP** outperforms the baselines with less training and inference costs highlights the significance of utilizing short soft prompts during training and inference. For example, **SMoP** saves 14.6% training time, 22.9% training memory, and 27.2% inference FLOPs in T5-large, compared to prompt tuning with a soft prompt of length 100. It is worth noting that full fine-tuning requires the fewest of FLOPs for inference as no additional tokens are added to the input, while **SMoP** introduces the least additional FLOPs.

### 3.2.2 Length and Number of Soft Prompts

To investigate the optimal length and number of soft prompts to employ, we present the experimental results on **SMoP** with diverse utilized prompt lengths and numbers of prompts in Table 2.

It is observed that increasing the total prompt length over 50 provides marginal performance gains. This finding is aligned with previous research (Lester et al., 2021; Li and Liang, 2021; Ma et al., 2022) that report increasing soft prompt length above a certain threshold brings limited improvements to performance.

Furthermore, we notice that using 20 soft prompts generally lead to a degradation in perfor-

---

[2]Measured with a single NVIDIA RTX A6000 GPU.

mance. We conjecture that this may be due to the limited labeled data for training in several Super-GLUE tasks, leading to insufficient training of each soft prompt (Wang et al., 2022).

Given these findings, we primarily report the results of **SMoP** utilizing 4 soft prompts, each with a length of 5 tokens. Note that while **SMoP** generally demonstrates improvements in prompt tuning, the optimal length and number of soft prompts may vary by specific tasks or datasets.

### 3.2.3 Routing Methods

To verify the impact of the routing method in the gating mechanism of **SMoP**, we perform experiments on diverse routing methods, including linear router without router perturbation (w/o perturbation), taking the weighted sum of two prompts with the highest probability (Top-2), Gumbel-Softmax routing where the output probability of the router is calculated as 1 (Gumbel-Softmax), stochastic routing (Stochastic) which is an application of AdaMix to prompt tuning (Zuo et al., 2022; Wang et al., 2022), and no routing (Single) which is identical to prompt tuning with a length 5 soft prompt.

Table 3 shows experimental results on three SuperGLUE tasks with diverse routing methods. The top-1 linear router with router perturbation, which is our original setting, generally outperforms all other routing strategies. One exception is BoolQ where removing the router perturbation exhibits a slightly better performance. We speculate that in high-resource settings like BoolQ, router perturbation may not be mandatory for sufficient training of each soft prompt.

## 4 Related Work

### 4.1 Prompt Tuning

Pre-trained language models (PLMs) have demonstrated remarkable performance on a wide range of tasks in Natural Language Processing (NLP) (Devlin et al., 2019; Liu et al., 2019). However, with the introduction of larger language models

such as T5 (Raffel et al., 2020) and GPT-3 (Brown et al., 2020), fine-tuning the entire parameters of the PLM for each specific task has become notably inefficient in terms of training and deployment.

To address such inefficiency, researchers have proposed parameter-efficient fine-tuning methods (Houlsby et al., 2019; Lester et al., 2021; Pfeiffer et al., 2021; Hu et al., 2022), which involves fine-tuning a relatively small portion of task-specific parameters of the PLM while keeping the other parameters frozen. Among these methods, prompt tuning (Lester et al., 2021) is a simple and effective approach that entails prepending learnable token embeddings (i.e., soft prompts) to the model input and solely fine-tuning these embeddings. The simplicity and adaptability of prompt tuning have led to several advancements aimed at improving its efficiency and performance by modifying the structure of soft prompts (Liu et al., 2021; Li and Liang, 2021), using instance-specific prompts (Jiang et al., 2022; Wu et al., 2022), or adjusting the training process (Huang et al., 2022; Ma et al., 2022). Moreover, prompt tuning is known for its capability for task knowledge transfer from source task prompts to target task prompts (Vu et al., 2022; Asai et al., 2022; Wang et al., 2023). These methods have improved the overall performance of prompt tuning, but they have overlooked the inefficiency of utilizing lengthy soft prompts. **SMoP** is designed to alleviate this efficiency concern and is orthogonal to most of the existing prompt tuning methods.

### 4.2 Mixture-of-Experts

Mixture-of-Experts is a model structure in which the output of the model is computed by multiple sub-networks (i.e., experts) conditionally activated by a gating mechanism (Shazeer et al., 2017). This enables increasing the number of model parameters without incurring a proportional increase in computation. Typically, the gating mechanism determines which experts process specific tokens (Shazeer et al., 2017; Fedus et al., 2022), while it can be extended to route sequences or batches (Wang et al., 2022; Zuo et al., 2022; Pan et al., 2023). In particular, Fedus et al. (2022) presents Switch Transformer that employs the Sparsely-Gated Mixture-of-Experts layer (Shazeer et al., 2017), and Zuo et al. (2022) proposes THOR which utilizes stochastic (i.e., random) routing.

Recently, Wang et al. (2022) has proposed AdaMix, a parameter-efficient fine-tuning method that integrates the concept of Mixture-of-Experts to Adapter (Houlsby et al., 2019). It follows THOR (Zuo et al., 2022) to employ stochastic routing and merging of multiple adapter modules. Both **SMoP** and AdaMix have taken inspiration from the concept of the Mixture-of-Experts structure to improve parameter-efficient fine-tuning. However, their primary motivations are distinct in that the motivation of **SMoP** is to use multiple short soft prompts for efficient prompt tuning, while the motivation of AdaMix is to provide multiple views of the given task for better performance. Therefore, **SMoP** employs a linear router for instance-wise prompt selection resulting in multiple soft prompts each specialized in a subset of the task, whereas AdaMix employs stochastic routing and merging, resulting in a single adapter module per task.

## 5 Conclusion

We have presented **SMoP** (**S**parse **M**ixture-**o**f-**P**rompts), a novel prompt tuning method that utilizes short soft prompts for efficient training and inference while maintaining performance gains associated with increased prompt length. To achieve this, we have employed a gating mechanism in **SMoP** that routes each instance to one of the multiple short soft prompts. Experimental results have demonstrated that **SMoP** has outperformed prompt tuning while reducing training and inference costs through the utilization of short soft prompts.

### Limitations

Given the same total prompt length, the gating mechanism of **SMoP** introduces additional parameters compared to prompt tuning, inducing additional storage requirements. Comparing prompt tuning with a soft prompt of length 20 (20,480 trainable parameters) and **SMoP** with 4 prompts of length 5 (24,576 trainable parameters) on T5-base, **SMoP** adds 20% trainable parameters and such difference increases as more prompts are utilized.

We further note that **SMoP** is orthogonal to most of the existing prompt tuning methods including prompt transfer learning methods (Vu et al., 2022; Asai et al., 2022; Wang et al., 2023) as mentioned in Section 4. While our investigation has highlighted the significance of incorporating short soft prompts through sparse activation in conventional single-task prompt tuning, we believe that **SMoP** holds promise as a valuable direction for augmenting the efficiency of prompt tuning methods in the future.

## Acknowledgements

We thank the anonymous reviewers for their helpful comments. This work was supported by the Basic Research Program through the National Research Foundation of Korea (NRF) grant funded by the Korea government (MSIT) (2021R1A2C3010430) and Institute of Information & Communications Technology Planning & Evaluation (IITP) grant funded by the Korea government (MSIT) (No.2019-0-00079, Artificial Intelligence Graduate School Program (Korea University)).

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

# Appendix

## A  Comparison to Adapter-based Methods

To further explore the advantages of **SMoP** in the realm of parameter-efficient fine-tuning methods, we compare **SMoP** and prompt tuning methods to adapter-based parameter-efficient fine-tuning methods, namely Adapter (Houlsby et al., 2019), AdapterFusion (Pfeiffer et al., 2021), and LoRA (Hu et al., 2022). We provide a brief description of each method and present the experimental results on six SuperGLUE tasks with the T5-base model.

Adapter-based methods add additional modules to the internal structure of the model. Adapter (Houlsby et al., 2019) adds bottleneck modules after the multi-head attention and feed-forward layer of each Transformer layer, while AdapterFusion (Pfeiffer et al., 2021) adds bottleneck modules only after the feed-forward layer. LoRA (Hu et al., 2022) adds a low-rank decomposition of each of the attention matrices, which are directly added during inference. We implement these methods upon the adapter-transformers[3] library.

Table 5 presents the experimental results of full fine-tuning, adapter-based methods, and prompt tuning methods on six SuperGLUE tasks with the T5-base model. While adapter-based methods generally outperform prompt tuning methods under their best configuration, **SMoP** is able to reach comparable performance while only utilizing up to $190\times$ a smaller number of trainable parameters. In particular, when the ratio of trainable parameters narrows to a factor of $33\times$, **SMoP** outperforms Adapter on 5 tasks out of 6. Similar results are observed for AdapterFusion, where **SMoP** shows inferior performance when the bottleneck dimension $d$ is set to 48, but reverses the results when $d$ is reduced to 8.

Considering LoRA, **SMoP** shows slightly better performance compared to both configurations. One notable result is that using a lower rank in LoRA does not yield a significant decrease in performance. However, as shown in Table 4, the level of parameter efficiency of **SMoP** is not attainable with LoRA, as LoRA ($r$=1) still requires $6\times$ more trainable parameters compared to **SMoP**. These observations highlight the parameter efficiency of **SMoP** compared to adapter-based approaches.

In general, adapter-based lightweight methods require additional parameters proportional to the

---

[3]https://github.com/adapter-hub/adapter-transformers

number of layers in the backbone model, as they add an adapter module to the internal structure of the original model. In contrast, prompt tuning methods including **SMoP** introduce additional parameters exclusively to the inputs of the model, enabling a parameter-efficient module where the number of trainable parameters doesn't increase proportionally to model size (Asai et al., 2022).

| Model | Method | Trainable Params % |
|---|---|---|
| T5-base | **SMoP** ($l$=5, $k$=4) | **0.0083 (1.0$\times$)** |
| | LoRA ($r$=1) | 0.0496 (6.0$\times$) |
| | LoRA ($r$=2) | 0.0991 (12.0$\times$) |
| | LoRA ($r$=4) | 0.1981 (24.0$\times$) |
| | LoRA ($r$=8) | 0.3954 (47.8$\times$) |

Table 4: Comparison of trainable parameter ratio between **SMoP** and LoRA. The value in the parenthesis for trainable params % denotes the relative difference, with **SMoP** as the reference point.

## B  Text-to-Text Templates

We provide the text-to-text templates and verbalizers used in our experiments in Table 6.

## C  Hyperparameters

We train our model for {50, 100} epochs on CB, COPA, RTE, WiC and for {10, 20} epochs on BoolQ and MultiRC with batch size 32, learning rate of {1e-4, 5e-5, 1e-5} for full fine-tuning and adapter-based methods, and learning rate {0.5, 0.3, 0.1} for prompt tuning methods including **SMoP**. We perform early stopping based on validation performance, and terminate training if there is no improvement for 10 epochs. We train the model with Adafactor optimizer (Shazeer and Stern, 2018), where the weight decay is 1e-5, and linear learning rate decay of warmup ratio 0.06 is applied.

## D  Detailed Experimental Results

We provide task-wise results of experiments presented in the paper. Since we experiment with our own train/validation/test split, the results may vary with previous works such as Lester et al. (2021).

### D.1  Performance

Table 7 and 8 present the experimental results on six SuperGLUE tasks on T5-base and T5-large.

### D.2  Training Costs

Table 9 presents the memory used during training (GB), and Table 10 presents the training time (s/100

| Model | Method | Trainable Params % | BoolQ %Acc | CB %Acc | COPA %Acc | MultiRC %F1$_a$ | RTE %Acc | WiC %Acc | Average Score (%) |
|---|---|---|---|---|---|---|---|---|---|
| | Full Fine-tuning | 100 | $81.9_{0.1}$ | $96.4_{1.8}$ | $64.3_{1.5}$ | $80.2_{0.2}$ | $79.2_{0.2}$ | $67.0_{2.3}$ | $78.2_{1.3}$ |
| | Adapter ($d$=48) | 1.5800 | $\mathbf{81.1}_{0.1}$ | $\mathbf{94.6}_{1.8}$ | $62.7_{0.6}$ | $\mathbf{80.2}_{0.2}$ | $76.6_{1.6}$ | $66.2_{1.6}$ | $76.9_{1.2}$ |
| | Adapter ($d$=8) | 0.2806 | $79.6_{0.6}$ | $89.9_{1.0}$ | $59.0_{1.0}$ | $80.1_{0.2}$ | $75.2_{1.5}$ | $65.3_{0.8}$ | $74.8_{0.9}$ |
| T5- | AdapterFusion ($d$=48) | 0.7963 | $79.2_{0.4}$ | $\mathbf{94.6}_{1.8}$ | $\mathbf{63.7}_{2.5}$ | $80.2_{0.2}$ | $\mathbf{79.2}_{0.9}$ | $66.8_{0.3}$ | $\mathbf{77.3}_{1.3}$ |
| base | AdapterFusion ($d$=8) | 0.1405 | $79.6_{0.6}$ | $92.3_{3.7}$ | $58.3_{0.6}$ | $79.9_{0.5}$ | $78.0_{2.9}$ | $64.9_{0.5}$ | $75.5_{2.0}$ |
| | LoRA ($r$=8) | 0.3954 | $79.0_{0.0}$ | $90.5_{1.0}$ | $60.0_{0.6}$ | $80.0_{0.0}$ | $77.9_{2.9}$ | $\mathbf{66.9}_{0.8}$ | $75.7_{1.3}$ |
| | LoRA ($r$=2) | 0.0991 | $79.1_{0.0}$ | $91.1_{0.0}$ | $59.3_{1.2}$ | $\mathbf{80.2}_{0.0}$ | $77.4_{2.7}$ | $66.5_{0.3}$ | $75.6_{1.2}$ |
| | Prompt Tuning ($l$=100) | 0.0344 | $79.1_{0.1}$ | $86.9_{3.7}$ | $56.7_{2.1}$ | $78.3_{0.2}$ | $73.2_{1.7}$ | $65.6_{1.2}$ | $73.3_{1.9}$ |
| | P-Tuning ($l$=20) | 0.1028 | $78.7_{0.2}$ | $91.7_{2.7}$ | $\mathbf{58.3}_{3.8}$ | $79.3_{0.2}$ | $77.3_{1.8}$ | $\mathbf{65.9}_{0.7}$ | $75.2_{2.1}$ |
| | **SMoP** ($l$=5, $k$=4) | 0.0083 | $\mathbf{79.4}_{0.3}$ | $\mathbf{94.6}_{1.8}$ | $58.3_{2.9}$ | $\mathbf{79.6}_{0.1}$ | $\mathbf{77.5}_{3.2}$ | $65.2_{0.5}$ | $\mathbf{75.8}_{1.9}$ |

Table 5: Experimental results of diverse parameter-efficient fine-tuning methods on six SuperGLUE tasks with T5-base model. The methods include full fine-tuning, adapter-based methods, prompt tuning methods, and our proposed **SMoP**. $d$ for Adapter and AdapterFusion indicates the bottleneck dimension and $r$ for LoRA indicates the rank of the matrices. The best performance among adapter-based methods and prompt tuning methods for each task are **bold** highlighted.

| Dataset | Text-to-text Template | Verbalizer |
|---|---|---|
| BoolQ | boolq passage: **passage** question: **question** | False, True |
| CB | cb hypothesis: **hypothesis**. premise: **premise** | entailment, contradiction, neutral |
| COPA | copa choice1: **choice1** choice2: **choice2** premise: **premise** question: **question** | choice1, choice2 |
| MultiRC | multirc question: **question** answer: **answer**. paragraph: **paragraph** | False, True |
| RTE | rte sentence1: **premise** sentence2: **hypothesis** | entailment, not_entailment |
| WiC | wic sentence1: **sentence1** sentence2: **sentence2** word: **word** | False, True |

Table 6: Text-to-text templates and verbalizers used in our experiments.

| Model | Method | Total Prompt Length | Utilized Prompt Length | BoolQ %Acc | CB %Acc | COPA %Acc | MultiRC %F1$_a$ | RTE %Acc | WiC %Acc | Average Score (%) |
|---|---|---|---|---|---|---|---|---|---|---|
| | Full Fine-tuning | - | - | $81.9_{0.1}$ | $96.4_{1.8}$ | $64.3_{1.5}$ | $80.2_{0.2}$ | $79.2_{0.2}$ | $67.0_{2.3}$ | $78.2_{1.3}$ |
| | | 5 | 5 | $78.5_{0.0}$ | $89.3_{1.8}$ | $54.0_{3.6}$ | $79.1_{0.1}$ | $69.9_{0.8}$ | $64.4_{0.0}$ | $72.5_{1.7}$ |
| | Prompt Tuning | 20 | 20 | $78.6_{0.0}$ | $86.9_{2.1}$ | $55.0_{3.5}$ | $79.2_{0.2}$ | $70.6_{1.8}$ | $64.3_{0.2}$ | $72.4_{1.8}$ |
| | | 50 | 50 | $79.3_{0.1}$ | $87.5_{1.8}$ | $56.0_{4.0}$ | $78.3_{0.0}$ | $70.8_{0.5}$ | $65.1_{0.2}$ | $72.8_{1.8}$ |
| | | 100 | 100 | $79.1_{0.1}$ | $86.9_{3.7}$ | $56.7_{2.1}$ | $78.3_{0.2}$ | $73.2_{1.7}$ | $65.6_{1.2}$ | $73.3_{1.9}$ |
| | | 5 | 5 | $79.0_{0.1}$ | $89.9_{3.7}$ | $59.0_{1.0}$ | $79.2_{0.1}$ | $73.8_{1.4}$ | $65.4_{1.3}$ | $74.4_{1.8}$ |
| | P-tuning | 20 | 20 | $78.7_{0.2}$ | $91.7_{2.7}$ | $58.3_{3.8}$ | $79.3_{0.2}$ | $77.3_{1.8}$ | $65.9_{0.7}$ | $75.2_{2.1}$ |
| | | 50 | 50 | $78.8_{0.2}$ | $90.5_{1.0}$ | $59.0_{1.0}$ | $79.2_{0.0}$ | $75.1_{1.6}$ | $65.2_{0.5}$ | $74.6_{0.9}$ |
| | | 100 | 100 | $79.0_{0.1}$ | $89.9_{1.0}$ | $59.0_{2.0}$ | $79.2_{0.0}$ | $73.8_{3.5}$ | $65.4_{0.7}$ | $74.4_{1.7}$ |
| | | 2 | 1 | $79.3_{0.3}$ | $90.7_{0.7}$ | $52.7_{4.2}$ | $78.8_{0.3}$ | $71.5_{2.5}$ | $64.7_{1.1}$ | $72.9_{2.1}$ |
| | | 4 | 1 | $79.0_{0.1}$ | $91.1_{1.8}$ | $57.3_{3.2}$ | $79.4_{0.1}$ | $72.4_{0.8}$ | $65.0_{0.4}$ | $74.0_{1.5}$ |
| | | 10 | 1 | $78.6_{0.0}$ | $92.9_{0.0}$ | $54.7_{1.2}$ | $78.9_{0.1}$ | $71.5_{3.0}$ | $64.3_{0.5}$ | $73.5_{1.3}$ |
| | | 20 | 1 | $78.6_{0.1}$ | $90.5_{1.0}$ | $57.7_{2.1}$ | $79.3_{0.2}$ | $72.6_{1.5}$ | $64.9_{0.8}$ | $73.9_{1.2}$ |
| | | 6 | 3 | $78.8_{0.1}$ | $92.9_{1.8}$ | $54.0_{5.0}$ | $79.1_{0.1}$ | $75.7_{1.8}$ | $64.7_{1.1}$ | $74.2_{2.3}$ |
| T5- | | 12 | 3 | $79.0_{0.2}$ | $92.9_{1.8}$ | $53.3_{5.5}$ | $79.2_{0.1}$ | $74.6_{0.2}$ | $64.9_{1.3}$ | $74.0_{2.4}$ |
| base | | 30 | 3 | $78.8_{0.1}$ | $94.0_{4.5}$ | $56.0_{3.6}$ | $79.2_{0.3}$ | $75.5_{0.5}$ | $65.6_{0.2}$ | $74.8_{2.4}$ |
| | | 60 | 3 | $78.7_{0.0}$ | $91.7_{1.0}$ | $56.0_{3.6}$ | $79.2_{0.1}$ | $74.7_{1.6}$ | $64.7_{0.1}$ | $74.2_{1.7}$ |
| | **SMoP** | 10 | 5 | $78.5_{0.0}$ | $92.9_{0.0}$ | $58.0_{4.6}$ | $79.4_{0.0}$ | $76.4_{1.3}$ | $64.9_{0.8}$ | $75.0_{2.0}$ |
| | | 20 | 5 | $79.4_{0.3}$ | $94.6_{1.8}$ | $58.3_{2.9}$ | $79.6_{0.1}$ | $77.5_{3.2}$ | $65.2_{0.5}$ | $\mathbf{75.8}_{1.9}$ |
| | | 50 | 5 | $79.3_{0.1}$ | $92.3_{1.0}$ | $58.7_{4.2}$ | $79.3_{0.0}$ | $77.1_{0.3}$ | $65.2_{0.4}$ | $75.3_{1.8}$ |
| | | 100 | 5 | $79.0_{0.3}$ | $93.4_{2.0}$ | $55.3_{3.1}$ | $79.3_{0.2}$ | $76.9_{2.0}$ | $64.3_{0.2}$ | $74.7_{1.7}$ |
| | | 20 | 10 | $78.7_{0.1}$ | $93.5_{1.0}$ | $59.3_{3.5}$ | $79.2_{0.3}$ | $76.0_{1.8}$ | $64.2_{0.1}$ | $75.1_{1.7}$ |
| | | 40 | 10 | $78.6_{0.1}$ | $92.3_{3.7}$ | $56.0_{1.7}$ | $78.9_{0.1}$ | $76.9_{0.0}$ | $66.4_{0.8}$ | $74.8_{1.7}$ |
| | | 100 | 10 | $78.5_{0.1}$ | $95.8_{1.0}$ | $57.7_{2.5}$ | $79.2_{0.1}$ | $75.1_{1.0}$ | $64.8_{1.7}$ | $75.2_{1.4}$ |
| | | 200 | 10 | $79.0_{0.4}$ | $91.1_{1.8}$ | $56.0_{3.5}$ | $79.4_{0.1}$ | $74.2_{2.8}$ | $64.9_{0.7}$ | $74.1_{2.0}$ |

Table 7: Experimental results on baseline methods and **SMoP** on six SuperGLUE tasks with T5-base. Subscripts of each score represent the corresponding standard deviation over multiple runs.

| Model | Method | Total Prompt Length | Utilized Prompt Length | BoolQ %Acc | CB %Acc | COPA %Acc | MultiRC %F1$_a$ | RTE %Acc | WiC %Acc | Average Score (%) |
|---|---|---|---|---|---|---|---|---|---|---|
| T5-large | Full Fine-tuning | - | - | $85.8_{0.1}$ | $96.4_{0.0}$ | $76.0_{2.6}$ | $84.5_{0.1}$ | $87.6_{0.4}$ | $70.3_{1.6}$ | $83.4_{1.3}$ |
| | Prompt Tuning | 5 | 5 | $83.3_{0.0}$ | $89.3_{2.5}$ | $57.5_{3.5}$ | $83.8_{0.0}$ | $86.3_{0.5}$ | $68.2_{0.2}$ | $78.1_{1.8}$ |
| | | 20 | 20 | $83.1_{0.1}$ | $90.2_{1.3}$ | $57.5_{6.4}$ | $83.8_{0.0}$ | $86.6_{0.0}$ | $68.5_{0.4}$ | $78.3_{2.7}$ |
| | | 50 | 50 | $83.1_{0.1}$ | $91.1_{0.0}$ | $58.5_{0.7}$ | $83.0_{0.1}$ | $85.9_{0.0}$ | $68.0_{0.4}$ | $78.2_{0.3}$ |
| | | 100 | 100 | $83.1_{0.2}$ | $90.5_{1.0}$ | $62.0_{3.0}$ | $82.6_{0.2}$ | $87.0_{1.0}$ | $66.2_{1.0}$ | $78.6_{1.4}$ |
| | P-Tuning | 5 | 5 | $83.2_{0.2}$ | $92.0_{3.7}$ | $69.0_{2.8}$ | $83.9_{0.1}$ | $86.6_{0.0}$ | $67.9_{1.1}$ | $80.4_{2.0}$ |
| | | 20 | 20 | $83.4_{0.4}$ | $91.7_{2.7}$ | $71.7_{3.2}$ | $84.2_{0.1}$ | $87.6_{1.0}$ | $69.2_{0.9}$ | $81.3_{1.8}$ |
| | | 50 | 50 | $83.5_{0.1}$ | $92.0_{1.3}$ | $71.0_{4.2}$ | $83.7_{0.0}$ | $87.0_{1.0}$ | $67.2_{0.8}$ | $80.7_{1.9}$ |
| | | 100 | 100 | $83.1_{0.1}$ | $93.8_{3.7}$ | $67.0_{1.4}$ | $82.2_{0.0}$ | $87.4_{0.5}$ | $65.7_{0.0}$ | $79.9_{1.6}$ |
| | SMoP | 10 | 5 | $83.0_{0.1}$ | $97.3_{1.3}$ | $69.5_{0.7}$ | $83.9_{0.0}$ | $86.6_{0.0}$ | $66.0_{0.4}$ | $81.1_{0.6}$ |
| | | 20 | 5 | $83.5_{0.1}$ | $96.4_{0.0}$ | $71.7_{3.1}$ | $83.9_{0.2}$ | $87.7_{0.0}$ | $68.6_{0.7}$ | $\mathbf{82.0}_{1.3}$ |
| | | 50 | 5 | $83.1_{0.1}$ | $94.7_{2.5}$ | $69.0_{4.2}$ | $83.9_{0.1}$ | $86.6_{2.5}$ | $68.2_{0.2}$ | $80.9_{2.3}$ |
| | | 100 | 5 | $83.6_{0.3}$ | $92.0_{1.3}$ | $67.5_{6.4}$ | $83.9_{0.1}$ | $88.8_{0.6}$ | $69.7_{0.6}$ | $80.9_{2.7}$ |

Table 8: Experimental results on baseline methods and **SMoP** on six SuperGLUE tasks with T5-large. Subscripts of each score represent the standard deviation over multiple runs.

steps) for each SuperGLUE task. For BoolQ and MultiRC in T5-large, we report the results for step batch size of 16 with gradient accumulation, as using batch size 32 exceeds the memory capacity of a single NVIDIA RTX A6000 GPU.

## D.3 Inference Costs

Table 11 presents the inference FLOPs (GFLOPs/sample) for each SuperGLUE task.

| Model | Method | Total Prompt Length | Utilized Prompt Length | BoolQ | CB | COPA | MultiRC | RTE | WiC | Average |
|---|---|---|---|---|---|---|---|---|---|---|
| T5-base | Full | - | - | 27.0 | 14.3 | 3.1 | 27.0 | 13.9 | 4.1 | 14.9 |
| | Prompt Tuning | 100 | 100 | 21.8 | 16.0 | 5.0 | 21.8 | 15.6 | 6.2 | 14.4 |
| | P-Tuning | 20 | 20 | 21.8 | 12.0 | 2.7 | 21.8 | 11.7 | 3.5 | 12.3 |
| | **SMoP** | 5 | 5 | 21.8 | 11.3 | 2.3 | 21.8 | 11.0 | 3.1 | 11.9 |
| T5-large | Full | - | - | 39.7 | 38.3 | 8.6 | 40.1 | 37.2 | 11.3 | 29.2 |
| | Prompt Tuning | 100 | 100 | 30.5 | 42.9 | 13.7 | 30.6 | 41.8 | 16.6 | 29.3 |
| | P-Tuning | 20 | 20 | 30.1 | 32.3 | 7.5 | 30.5 | 31.3 | 9.8 | 23.6 |
| | **SMoP** | 5 | 5 | 30.0 | 30.5 | 6.4 | 30.5 | 29.5 | 8.6 | 22.6 |

Table 9: Peak memory (GB) during training on SuperGLUE tasks.

| Model | Method | Total Prompt Length | Utilized Prompt Length | BoolQ | CB | COPA | MultiRC | RTE | WiC | Average |
|---|---|---|---|---|---|---|---|---|---|---|
| T5-base | Full | - | - | 105.8 | 92.6 | 45.8 | 131.6 | 76.5 | 36.0 | 81.4 |
| | Prompt Tuning | 100 | 100 | 93.1 | 90.3 | 37.2 | 103.7 | 71.4 | 28.4 | 70.7 |
| | P-Tuning | 20 | 20 | 84.8 | 85.9 | 30.5 | 108.2 | 59.0 | 21.1 | 64.9 |
| | **SMoP** | 5 | 5 | 82.5 | 74.1 | 30.8 | 104.6 | 54.2 | 19.8 | 61.0 |
| T5-large | Full | - | - | 228.4 | 183.1 | 82.8 | 338.9 | 152.0 | 71.3 | 176.1 |
| | Prompt Tuning | 100 | 100 | 204.3 | 169.1 | 74.9 | 253.0 | 137.6 | 68.3 | 151.2 |
| | P-Tuning | 20 | 20 | 171.2 | 134.9 | 51.5 | 275.9 | 114.0 | 43.7 | 131.9 |
| | **SMoP** | 5 | 5 | 164.7 | 134.0 | 47.4 | 281.0 | 107.4 | 39.9 | 129.1 |

Table 10: Training time (s/100 steps) on SuperGLUE tasks.

| Model | Method | Total Prompt Length | Utilized Prompt Length | BoolQ | CB | COPA | MultiRC | RTE | WiC | Average |
|---|---|---|---|---|---|---|---|---|---|---|
| T5-base | Full | - | - | 119.4 | 86.7 | 13.8 | 105.7 | 78.3 | 17.6 | 70.3 |
| | Prompt Tuning | 100 | 100 | 136.9 | 120.2 | 40.1 | 130.7 | 114.3 | 46.7 | 98.1 |
| | P-Tuning | 20 | 20 | 124.3 | 93.4 | 19.0 | 114.1 | 85.5 | 23.4 | 76.6 |
| | **SMoP** | 5 | 5 | 119.2 | 88.4 | 15.1 | 107.3 | 80.1 | 19.0 | 71.5 |
| T5-large | Full | - | - | 402.5 | 334.9 | 48.4 | 365.1 | 274.5 | 61.4 | 247.8 |
| | Prompt Tuning | 100 | 100 | 507.3 | 633.5 | 141.0 | 421.7 | 400.7 | 164.2 | 378.1 |
| | P-Tuning | 20 | 20 | 417.6 | 499.2 | 66.8 | 384.5 | 299.6 | 81.8 | 291.6 |
| | **SMoP** | 5 | 5 | 406.3 | 474.1 | 53.0 | 371.7 | 280.8 | 66.5 | 275.4 |

Table 11: Inference FLOPs (GFLOPs/sample) on SuperGLUE tasks.