# OpenReview forum: "SMoP: Towards Efficient and Effective Prompt Tuning with Sparse Mixture-of-Prompts"
_EMNLP/2023/Conference — EMNLP 2023 Main_

### Official Review · Reviewer_xer5 · 2023-07-31

**Soundness:** 3

**Excitement:**

3: Ambivalent: It has merits (e.g., it reports state-of-the-art results, the idea is nice), but there are key weaknesses (e.g., it describes incremental work), and it can significantly benefit from another round of revision. However, I won't object to accepting it if my co-reviewers champion it.

**Paper Topic And Main Contributions:**

This paper proposes SMoP, a mixture-of-expert method to approach parameter-efficient fine-tuning (PEFT). The authors first note that previous literature has noted the improved performance of using longer soft prompts over short ones, but doing so increases the context length and thus increases the computational expense. SMoP instead tunes multiple short prompts but combines them using a gating mechanism, a network that selects and weighs the different short prompts that are trained jointly. The authors performed experiments on two T5 models and show improvement in cost-performance trade-off compared to previous prompt tuning methods in a suite of SuperGLUE datasets.

**Questions For The Authors:**

Please address the concerns in *Reasons to Reject*.

I view the first point of *Reason to Reject* as the biggest concern and I'll adjust my rating conditional on the other reviews and if the authors could provide a satisfactory author response.

## Post-rebuttal comments

I thank the authors for providing a detailed rebuttal. Below, I provide detailed comments:
1. Comparison with AdaMix: Thank you for the clarification. Please incorporate the changes in the camera-ready version.
2. Comparison with LoRA: I do not disagree with the authors' saying that prompt tuning is less invasive, but my point was that prompt tuning while requiring no architecture changes, still requires *access* to the parameters, and you still cannot run prompt tuning in models where you may only observe their outputs in a black-box manner, and thus there are only limited scenarios where you can run prompt tuning but not, for example, LoRA -- I am stating this for clarification, not as a point against the authors as they indeed provide LoRa comparison. While I thank the authors for providing additional experiments in a short timeframe, my concerns are: 1) It seems that the difference between LoRA and SMoP is minimal and likely statistically insignificant, and 2) while SMoP is indeed more parameter-efficient in the Table, both LoRA and SMoP involve hyperparameters that significantly affect their parameter efficiency, and it is unknown whether a lower rank value in LoRA would lead to a similarly competitive performance (one key advantage claimed by LoRA authors is that LoRA is not very sensitive to the rank choice).

In general, I have become more positive about the work and would have increased the excitement score to 3.5 had I had the choice as in previous years. The reason I cannot give an even higher score is stated in Point 2 of my response.

**Reasons To Accept:**

- I think the overall methodology is reasonable and convincing and addresses an important problem (as the authors state on Page 1 that "they have overlooked the inefficiency" associated with longer soft prompts).
- As briefly mentioned in Limitations, I think the proposed method can be largely plug-and-play for various soft prompt tuning methods, as any prompt tuning methods utilising the soft tokens could be potentially broken up into small pieces with a routing network to achieve reduced cost. I think this is another plus in favour of this paper.
- The experiments are executed satisfactorily. The ablation studies are included, such as the trade-off between prompt length and number of prompts and the use of different routing mechanisms. I think these experiments would greatly improve readers' confidence in the method's working mechanism.

**Reasons To Reject:**

- One of my major concerns is that the relation to previous works should be more explicitly stated. In my view, the proposed method is very similar to the previous work of AdaMix. While AdaMix uses a mix of *adapters*, the present paper uses a mix of *prompts*. However, the soft prompts can be deemed as adapters only inserted into the embedding layer of the PLMs, and thus I'd be grateful if the authors could discuss especially w.r.t this very related work. While the paper did cite AdaMix (Wang et al. 2022), the only references are 1) that using many soft prompts leads to performance deterioration and 2) comparing against stochastic routing. I feel that the similarity (at least *prima facie*) at a more fundamental level to this work warrants a more thorough discussion.

- While the authors compare favourably against methods such as prompt tuning and P-tuning, I'm wondering how does the method compare to other lightweight methods that have been recently shown to be very effective, such as LoRA? In decomposing updates into low-rank matrices, LoRA similarly reduces the training cost by effectively reducing the number of parameters to fine-tune during the PEFT procedure, and it also offers an adjustable parameter (i.e., the rank of the update) that allows the user to trade-off the cost and precision. The authors tried to argue on Page 1 that prompt tuning is superior in its simplicity and that "it does not require modifications to the model structure" -- I think soft prompt tuning still requires modifying the model structure, just in a less invasive way; the input embedding layer still has to be modified (c.f. hard prompt tuning requiring only modification on the input texts).

- Another minor point related to the Strength I mentioned: while the authors conceptually argued for (and I agree with) that the proposed method is orthogonal to various prompt tuning methods, it would be much more convincing if the authors provided empirical evidence.  However, I do not view this point as a critical point against the authors but rather as a suggestion for future improvement.

**Reproducibility:**

4: Could mostly reproduce the results, but there may be some variation because of sample variance or minor variations in their interpretation of the protocol or method.

**Reviewer Confidence:**

4: Quite sure. I tried to check the important points carefully. It's unlikely, though conceivable, that I missed something that should affect my ratings.

---

> ### Author Rebuttal · Authors · 2023-08-29
>
> Thank you for your encouraging words and insightful suggestions. Below are our responses, and we are keen to address any further concerns.
>
> **Reasons to reject 1. Relation and discussion with respect to previous works (AdaMix) should be more explicitly stated.**
> - Thank you for your suggestion, we would like to provide an explicit comparison of SMoP and AdaMix, which will be included in the camera-ready version.
> - SMoP and AdaMix have substantial differences in their motivation and methodology. The primary motivation of SMoP is to use multiple short soft prompts for efficient prompt tuning, and each soft prompt is to be specialized into a subset of the task dataset. Therefore, SMoP employs a linear router for instance-wise prompt selection. Conversely, the motivation of AdaMix is to provide multiple views of the given task for better performance [1]. Hence, AdaMix employs stochastic routing and merging, resulting in a single adapter module per task.  Thus, while both works have taken inspiration from the concept of the Mixture-of-Experts structure, their motivations and methodologies are distinct.
>
> - Additionally, we have empirically observed that directly applying the single-module-per-task methodology of AdaMix to prompt tuning leads to lower performance compared to SMoP. Table 3 in the paper shows the results regarding this observation (SMoP 83.8 vs. Stochastic 78.1). To clarify, the “Stochastic” configuration in Table 3 mirrors the method of AdaMix, including consistency regularization and merging.
>
> **Reasons to reject 2. Clarification in the superiority of prompt tuning and comparison to other lightweight methods (e.g., LoRA).**
> - First, we would like to clarify the superiority of prompt tuning. Prompt tuning approaches modify the input of the model by prepending soft prompt tokens to the original input. This adjustment does not necessarily entail modification to the internal structure of the model [2].
> - A clear example that emphasizes the difference is where several tasks are processed in the same batch. Prompt tuning can handle this with a single model since different prompts corresponding to the task of each input can be prepended, but adapter-based approaches require copies of the model for each task [3].
> - Another advantage of prompt tuning approaches is their parameter efficiency. They solely introduce additional parameters to the input of the model without being influenced by the number of layers in the underlying backbone model.
> - As the main motivation of SMoP is to alleviate the inefficiencies in prompt tuning associated with long soft prompts, we have presented a comparison with prompt tuning methods in Table 1 as our main experimental results. To address your concerns regarding comparison to other lightweight methods, we conducted additional experiments with the T5-base model on three adapter-based approaches, namely Adapter [4], AdapterFusion [5], and LoRA [6]. The results are as follows:
>
> | Method | Trainable % | BoolQ | CB| COPA| MultiRC | RTE | WiC | Avg. |
> |:---:|:---:|:---:|:---:|:---:|:---:|:---:|:---:|:---:|
> | Adapter (m=48) | 1.5800 | 81.0 | 95.5 | 60.3 | 80.3 | 75.8 | 67.0 | **76.6** |
> | AdapterFusion (m=48) | 0.7963 | 79.5 | 94.6 | 58.3 | 79.2 | 79.2 | 65.9 | 76.1 |
> | LoRA (rank=8) | 0.3954 | 79.0 | 90.5 | 59.7 | 80.0 | 77.9 | 66.9 | 75.7 |
> | Adapter (m=8) | 0.2806 | 79.0 | 90.2 | 58.0 | 80.0 | 73.6 | 65.1 | 74.3 |
> | **SMoP (l=5, k=4)** | **0.0083** | 79.4 | 94.6 | 58.3 | 79.6 | 77.5 | 65.2 | 75.8 |
>
> - In this table, "m" for Adapters represents the bottleneck dimension of the adapter layer, whereas "l" and "k" for SMoP represent the utilized soft prompt length and the number of soft prompts. The methods are listed in descending order based on their trainable parameter ratio.
>
> - The experimental results demonstrate that SMoP achieves comparable performance to adapter-based lightweight methods while utilizing less than 47~190 times the trainable parameters. Notably, SMoP outperforms adapter with bottleneck dimension 8 which employs 33 times more trainable parameters than SMoP. These results highlight the parameter efficiency of SMoP compared to adapter-based approaches. We will include these results in our camera-ready version.
>
> **Reason to reject 3. Empirical evidence on orthogonality of SMoP to various prompt tuning methods.**
> - We believe that exploring the application of SMoP to various prompt tuning methods holds significant potential for future research, and we are engaged in conducting investigations in this direction. Thank you for your suggestion.
>
> References
>
> [1] Wang et al., "AdaMix: Mixture-of-Adaptations for Parameter-efficient Model Tuning", EMNLP 2022
>
> [2] Pfeiffer et al., “Modular Deep Learning”, arXiv 2023
>
> [3] Bari et al., “SPT: Semi-Parametric Prompt Tuning for Multitask Prompted Learning”, arXiv 2022
>
> [4] Houlsby et al., “Parameter-Efficient Transfer Learning for NLP”, ICML 2019
>
> [5] Pfeiffer et al., “AdapterFusion: Non-Destructive Task Composition for Transfer Learning”. EACL 2021
>
> [6] Hu et al., “LoRA: Low-Rank Adaptation of Large Language Models”, ICLR 2022

---

### Official Review · Reviewer_k8Ru · 2023-08-04

**Soundness:** 4

**Excitement:**

4: Strong: This paper deepens the understanding of some phenomenon or lowers the barriers to an existing research direction.

**Paper Topic And Main Contributions:**

This work focuses on improving the efficacy of prompts using short prompts using a sparse mixture. The idea is equivalent to a sparse mixture of models, and in a similar manner, it has a gating mechanism that allows the proposed model to specialize different prompts for different subsets of the dataset. The paper supports its claims with strong empirical performance without compromising the training and inference computation. Furthermore, this work uses memory efficiently and scales well with prompt length.

**Reasons To Accept:**

- This work presents a clear and well-motivated technique that requires less GPU memory. The overall idea is simple and is derived from past work on sparse mixture of experts models from the DL community. This makes the work accessible to a wider community and increases its applicability.
- Experiment results are satisfactory and validate the main premise of the paper, namely that SMoP is effective.



**Reasons To Reject:**

- Given the numbers are so close for both tables 1 and 3, it might be worth to compute their variance computed over different runs.

**Reproducibility:**

3: Could reproduce the results with some difficulty. The settings of parameters are underspecified or subjectively determined; the training/evaluation data are not widely available.

**Reviewer Confidence:**

4: Quite sure. I tried to check the important points carefully. It's unlikely, though conceivable, that I missed something that should affect my ratings.

---

> ### Author Rebuttal · Authors · 2023-08-29
>
> Thank you for your encouraging words and insightful suggestions. Below is our response, and we are keen to address any further concerns.
>
> **Reason to reject 1. Given the numbers are so close for both tables 1 and 3, it might be worth to compute their variance computed over different runs.**
>
> - We have conducted experiments on 3 different random seeds and reported the average value derived from these runs. The results along with their corresponding standard deviations are presented in the tables below.
>
> **Table 1**
> |   Model  |     Method    | $l$ ($k$) |   BoolQ   |     CB    | COPA | MultiRC | RTE | WiC | Avg. |
> |:--------:|:-------------:|:-------------:|:---------:|:---------:|:----:|:-------:|:---:|:---:|:----:|
> |  T5-base |      Full     |       - (-)       | $81.9_{0.1}$ | $96.4_{1.8}$ |   $64.3_{1.5}$  | $80.2_{0.2}$ | $79.2_{0.2}$ | $67.0_{2.3}$ | $78.2_{1.3}$    |
> |  | Prompt Tuning |      100 (-)      | $79.1_{0.1}$| $86.9_{3.7}$ |$56.7_{2.1}$|$78.3_{0.2}$|$73.2_{1.7}$|$65.6_{1.2}$|$73.3_{1.9}$|
> |  |    P-Tuning   |       20 (-)      |$78.7_{0.2}$|$91.7_{2.7}$|$58.3_{3.8}$|$79.3_{0.2}$|$77.3_{1.8}$|$65.9_{0.7}$|$75.2_{2.1}$|
> |  |      SMoP     |     5 (4)    |$79.4_{0.3}$|$94.6_{1.8}$|$58.3_{2.9}$|$79.6_{0.1}$|$77.5_{3.2}$|$65.2_{0.5}$|$75.8_{1.9}$|
> | T5-large |      Full     |       - (-)      |$85.8_{0.1}$|$96.4_{0.0}$|$76.0_{2.6}$|$84.5_{0.1}$|$87.6_{0.4}$|$70.3_{1.6}$|$83.4_{1.3}$|
> | | Prompt Tuning |      100 (-)      |$83.1_{0.2}$|$90.5_{1.0}$|$62.0_{3.0}$|$82.6_{0.2}$|$87.0_{1.0}$|$66.2_{1.0}$|$78.6_{1.4}$|
> |  |    P-Tuning   |       20 (-)       |$83.4_{0.4}$|$91.7_{2.7}$|$71.7_{3.2}$|$84.2_{0.1}$|$87.6_{1.0}$|$69.2_{0.9}$|$81.3_{1.8}$|
> | |      SMoP     |     5 (4)    |   $83.5_{0.1}$   |$96.4_{0.0}$|$71.7_{3.1}$|$83.9_{0.2}$|$87.7_{0.0}$|$68.6_{0.7}$|$82.0_{1.3}$|
>
> **Table 3**
> |  Model  |      Method      | BoolQ | CB | RTE | Avg. |
> |:-------:|:----------------:|:-----:|:--:|:---:|:----:|
> | T5-base |       SMoP       |$79.4_{0.3}$|$94.6_{1.8}$|$77.5_{3.2}$|$83.8_{2.1}$|
> |         | w/o perturbation |$79.7_{0.2}$|$93.5_{2.7}$|$76.0_{1.5}$|$83.1_{1.8}$|
> |         |       Top-2      |$78.4_{0.2}$|$88.1_{1.0}$|$69.7_{0.4}$|$78.7_{0.6}$|
> |         |  Gumbel-Softmax  |$79.2_{0.4}$|$92.3_{2.0}$|$75.2_{4.3}$|$82.2_{2.7}$|
> |         |    Stochastic    |$78.2_{0.3}$|$86.9_{2.1}$|$69.2_{1.7}$|$78.1_{1.6}$|
>
> - We will include the standard deviations of our experimental results with the clarification on our experimental results. Thank you for your valuable comment.

---

### Official Review · Reviewer_qdfJ · 2023-08-04

**Soundness:** 3

**Excitement:**

2: Mediocre: This paper makes marginal contributions (vs non-contemporaneous work), so I would rather not see it in the conference.

**Missing References:**

1. Vu T, Lester B, Constant N, Al-Rfou’ R, Cer D. SPoT: Better Frozen Model Adaptation through Soft Prompt Transfer. ACL 2022.

2. Asai A, Salehi M, Peters ME, Hajishirzi H. ATTEMPT: Parameter-Efficient Multi-task Tuning via Attentional Mixtures of Soft Prompts. EMNLP 2022.

3. Wang Z, Panda R, Karlinsky L, Feris R, Sun H, Kim Y. Multitask Prompt Tuning Enables Parameter-Efficient Transfer Learning. ICLR 2023.

**Paper Topic And Main Contributions:**

This paper proposes a sparse mixture of soft prompts, SMoP, that alleviates the training and inference cost for long soft prompts. It implements a linear router model for assigning the soft prompt with the highest probability as the final prompt. In addition, it introduces a router perturbation trick to encourage the exploration of different prompts in the training phase.  By conducting experiments on T5-base and large on 6 SuperGLUE tasks, it shows improved training cost and marginal performance improvement over the Prompt-Tuning and P-tuning baselines. However, other important baselines that show strong similarity in principle: Attempt [1], SPoT [2], MPT [3], are neither discussed nor compared. Therefore, the soundness of this work needs to be further improved. Moreover, due to the strong overlapping between the proposed methodology of SMoP and Attempt, the novelty of this work is very limited. Though Attempt is designed to focus on multi-task prompt transfer learning, Attempt can be interpreted as an improved version of SMoP with an additional attention module, whereas SMoP only considers a single soft prompt with the highest probability.

**Questions For The Authors:**

1. Could you please explain how you derive the trainable parameters in Table 1, especially for SMoP?
2. Why do you use the soft prompt with the highest probability only instead of using a weighted average of all learned prompts? Have you considered the strategy used in AdaMix?
3. In your ablation study, can you provide a comparison to prompt-tuning with the same final length of prompts?

I appreciate if the author could respond to the concerns in the weakness section.

----------
Thanks for the response from the author. With more detailed experimental evidence, I increased my score for the solidness. I also appreciate the discussion from the author for the difference between SMoP and ATTEMPT. However, I still find the SMoP as a simplified and single-task version of ATTEMPT, which limits the whole novelty of this work, so I kept my excitement score unchanged. I believe a discussion between SMoP and relevant multi-task framework such as SPoT, Attempt, and MPT will be neccessary for making this work distinct for future manuscripts.

**Reasons To Accept:**

1. This method shows improved training and inference costs for the traditional prompt tuning.
2. It is particularly designed for long soft prompts and will be beneficial in that case.

**Reasons To Reject:**

1. Important baselines and critical comparisons are missing, including but not limited to: SPoT, Attempt, and MPT.
2. Limited novelty given the Attempt framework.
3. The proposed method requires further expensive hyper-parameter search, as shown in Table 2, and the optimal k, l setups are not easy to obtain, which makes the overall training more expensive and redundant than the original prompt tuning implementation.
4. The experiments do not report multiple random seeds, and it appears to me that the improved performance in Table 1 can be due to cherry picking the seed and the exhaustive hyper-parameter search in Table 2.
5. As described by the author in Limitation section, the SMoP method requires 20% more trainable parameters, but the reported training parameters in Table 1 is misleading?

**Reproducibility:**

4: Could mostly reproduce the results, but there may be some variation because of sample variance or minor variations in their interpretation of the protocol or method.

**Reviewer Confidence:**

4: Quite sure. I tried to check the important points carefully. It's unlikely, though conceivable, that I missed something that should affect my ratings.

---

> ### Author Rebuttal · Authors · 2023-08-29
>
> Thank you for your constructive comments and insightful suggestions. Below are our responses, and we are keen to address any further concerns.
>
> **Reason to reject 1. Important baselines and critical comparisons are missing, including but not limited to: SPoT, Attempt, and MPT.**
> - Works such as SPoT, ATTEMPT, and MPT are beyond the scope of our work. These works, encompassing multi-task prompt transfer learning, are centered on the transferability of task-specific knowledge from source tasks to a target task. In contrast, SMoP focuses on efficient single-task prompt tuning by alleviating the inefficiency associated with long soft prompts during the training.
> - In essence, multi-task transfer learning methods focus on verifying and utilizing well-initialized prompts via source tasks, while SMoP focuses on efficiently, and effectively initializing the prompts. Thus, we firmly believe that SMoP is rather orthogonal to multi-task prompt transfer learning and the application of SMoP to such work is a promising avenue for future research.
> - We would appreciate any further suggestions on additional baseline methods that will strengthen our paper.
>
> **Reason to reject 2. Limited novelty given the ATTEMPT framework.**
> - SMoP has several novel aspects compared to ATTEMPT.
> 1) Motivation: ATTEMPT’s motivation is to leverage information from multiple source tasks via soft prompts [1], while SMoP’s motivation is to alleviate the inefficiency of soft prompt tuning.
> 2) Methodology: The attention module of ATTEMPT assumes well-initialized soft prompts that are frozen during target prompt training. In contrast, SMoP's linear router does not necessitate initialized soft prompts, enabling simultaneous training of all soft prompts. This divergence leads to ATTEMPT's two-stage training process (source prompt training and target prompt training), while SMoP operates within a unified, single-stage approach.
> 3) Parameter-Efficiency: During training and inference, ATTEMPT with T5-base involves 96k parameters plus the source task prompts per target task, while SMoP with T5-base in our best-performing configuration employs <20k parameters per target task, therefore is highly parameter-efficient.
> - Due to these differences, we believe that it could be misleading to interpret ATTEMPT as an improved version of SMoP.
>
> **Reason to reject 3. The proposed method requires further expensive hyper-parameter search, as shown in Table 2, and the optimal k, l setups are not easy to obtain, which makes the overall training more expensive and redundant than the original prompt tuning implementation.**
>
> - While SMoP does involve managing two parameters associated with soft prompts and requires more experiments to identify optimal configurations, the enhancements offered by SMoP are not significantly impacted by the choice of hyperparameters. For example, as illustrated in Table 2 of the paper, as long as the utilized soft prompt length is 5, SMoP consistently outperforms prompt tuning with a soft prompt of length 100 regardless of the number of soft prompts (74.7 ~ 75.8 vs 73.3). Hence, the increased number of parameters does not necessarily lead to expensive training costs.
>
> **Reason to reject 4. The experiments do not report multiple random seeds, and it appears to me that the improved performance in Table 1 can be due to cherry picking the seed and the exhaustive hyper-parameter search in Table 2.**
> - To clarify, we have reported the average result from the experimental results of three distinct random seeds. The results along with their corresponding standard deviations of our main experiments are presented below:
>
> |   Model  |     Method    | $l$ ($k$) |   BoolQ   |     CB    | COPA | MultiRC | RTE | WiC | Avg. |
> |:--------:|:-------------:|:-------------:|:---------:|:---------:|:----:|:-------:|:---:|:---:|:----:|
> |  T5-base |      Full     |       - (-)       | $81.9_{0.1}$ | $96.4_{1.8}$ |   $64.3_{1.5}$  | $80.2_{0.2}$ | $79.2_{0.2}$ | $67.0_{2.3}$ | $78.2_{1.3}$    |
> |  | Prompt Tuning |      100 (-)      | $79.1_{0.1}$| $86.9_{3.7}$ |$56.7_{2.1}$|$78.3_{0.2}$|$73.2_{1.7}$|$65.6_{1.2}$|$73.3_{1.9}$|
> |  |    P-Tuning   |       20 (-)      |$78.7_{0.2}$|$91.7_{2.7}$|$58.3_{3.8}$|$79.3_{0.2}$|$77.3_{1.8}$|$65.9_{0.7}$|$75.2_{2.1}$|
> |  |      SMoP     |     5 (4)    |$79.4_{0.3}$|$94.6_{1.8}$|$58.3_{2.9}$|$79.6_{0.1}$|$77.5_{3.2}$|$65.2_{0.5}$|$75.8_{1.9}$|
> | T5-large |      Full     |       - (-)      |$85.8_{0.1}$|$96.4_{0.0}$|$76.0_{2.6}$|$84.5_{0.1}$|$87.6_{0.4}$|$70.3_{1.6}$|$83.4_{1.3}$|
> | | Prompt Tuning |      100 (-)      |$83.1_{0.2}$|$90.5_{1.0}$|$62.0_{3.0}$|$82.6_{0.2}$|$87.0_{1.0}$|$66.2_{1.0}$|$78.6_{1.4}$|
> |  |    P-Tuning   |       20 (-)       |$83.4_{0.4}$|$91.7_{2.7}$|$71.7_{3.2}$|$84.2_{0.1}$|$87.6_{1.0}$|$69.2_{0.9}$|$81.3_{1.8}$|
> | |      SMoP     |     5 (4)    |   $83.5_{0.1}$   |$96.4_{0.0}$|$71.7_{3.1}$|$83.9_{0.2}$|$87.7_{0.0}$|$68.6_{0.7}$|$82.0_{1.3}$|
>
> - Furthermore, as stated in Appendix B, we have conducted a hyperparameter search on the baseline methods consistently with SMoP, which include learning rate, batch size, and training epochs.
> - We will report the experimental setup and results regarding the standard deviations in our paper.
>
> **Reason to reject 5. As described by the author in Limitation section, the SMoP method requires 20% more trainable parameters, but the reported training parameters in Table 1 is misleading?**
> - The setting in the limitation is where the total number of prompts is the same (a single soft prompt of length 20 for prompt tuning and four soft prompts of length 5 for SMoP). In contrast, the settings outlined in Table 1 are the configurations that yield the best performance (a single soft prompt of length 100 for prompt tuning and four soft prompts of length 5 for SMoP).
> - For the first case where the total number of prompts is identical, the linear router in SMoP introduces an additional parameter to tune given the same total number of prompts. The specific numbers are 15,360 vs. 18,432 trainable parameters for prompt tuning and SMoP, respectively.
> - For the second case where SMoP utilizes fewer soft prompts, the number of trainable parameters is lower than prompt tuning. The specific numbers are 76,800 vs. 18,432 trainable parameters for prompt tuning and SMoP, respectively.
>
> **Question 1. Could you please explain how you derive the trainable parameters in Table 1, especially for SMoP?**
> - The number of trainable parameters in prompt tuning can be derived as a sum of two components: a) soft prompt embeddings and b) additional parameters to reparametrize/select soft prompt embeddings.
> - Based on this, we provide our calculation process of trainable parameters in Table 1 on T5-base as an example. For T5-base, the embedding dimension e is 768, and the total count of pre-trained parameters is 222,882,048.
> 1. SMoP
> - a) Uses four soft prompts of length 5, summing up to a total prompt length of 20. This counts up to 20 * 768 = 15,360 parameters.
> - b) The linear router for prompt selection is of shape e * number of prompts, which is 4 * 768 = 3,072 parameters.
> - The total number of trainable parameters is 15,360 + 3,072 = 18,432 parameters.
> - The ratio is computed as 18,432 / (18,432 + 222,882,048) * 100, which is approximately 0.0083%.
> 2. Prompt Tuning
> - a) Uses a single soft prompt of length 100, which counts up to 100 * 768 = 76,800 parameters.
> - b) There are no additional parameters in Prompt Tuning.
> - The total number of trainable parameters is 76,800 parameters.
> - The ratio is computed as 76,800 / (76,800 + 222,882,048) * 100, which is approximately 0.0344%.
> 3. P-tuning
> - a) Uses a single soft prompt of length 20, which counts up to 20 * 768 = 15,360 parameters.
> - b) A 3-layer MLP with the width of each FC layer 128, 128, and 768 is used for reparameterization. The number of parameters in each layer is computed as 768 * 128 + 128, 128 * 128 + 128, and 128 * 768 + 768. This sums up to 212,992 parameters.
> - The total number of trainable parameters is 15,360 + 212,992 = 228,352 parameters
> - The ratio is computed as 229,376 / (229,376 + 222,882,048) * 100, which is approximately 0.1028%.
>
> **Question 2. Why do you use the soft prompt with the highest probability only instead of using a weighted average of all learned prompts? Have you considered the strategy used in AdaMix?**
> - To clarify, the ‘Stochastic’ setting in Table 3 of our paper mirrors the methodology of AdaMix. Also, we have considered a weighted sum of soft prompts with top-2 probability, which is denoted as ‘Top-2’ in Table 3. We have observed that both methods underperform SMoP (SMoP 83.8 vs Top-2 78.7 vs Stochastic 78.1). As we have observed that the weighted sum of 2 soft prompts harms performance, we did not conduct any further experiments on the weighted sum of learned soft prompts.
> - We conjecture that during training, each soft prompt becomes an expert to the subset of the data, and the instance-wise selection of a single prompt is advantageous over the approach of combining multiple prompts in a mixture. This is in line with previous literature that highlights the significance of instance-specific soft prompts [2, 3].
>
> **Question 3. In your ablation study, can you provide a comparison to prompt-tuning with the same final length of prompts?**
> - Regarding the final length of prompts as the utilized prompt length, we provide a comparison between prompt tuning with soft prompt length 5 and SMoP with four length 5 soft prompts as an example in the table below.
>
> | Model | Method | Total Prompt Length | Utilized Prompt Length | Trainable Params % | Training Time | Memory Used | Inference FLOPs | SuperGLUE Average |
> |---|---|---|---|---|---|---|---|---|
> | T5-base | Prompt Tuning | 5 | 5 | 0.0017 | 60.7 | 11.9 | 71.5 | 72.5 |
> |  | SMoP | 20 | 5 | 0.0083 | 61.0 | 11.9 | 71.5 | 75.8 |
> | T5-large | Prompt Tuning | 5 | 5 | 0.0007 | 123.5 | 22.6 | 275.4 | 78.1 |
> |  | SMoP | 20 | 5 | 0.0033 | 129.1 | 22.6 | 275.4 | 82.0 |
>
> - To clarify, the memory usage and inference FLOPs are not identical, but the increased cost ratio is lower than 1e-6 which is negligible and thus not presented in the table.
> - We will include this comparison in our work as an ablation study as ‘single’.
>
> References
>
> [1] Asai et al., "ATTEMPT: Parameter-Efficient Multi-task Tuning via Attentional Mixtures of Soft Prompts", EMNLP 2022
>
> [2] Wu et al., "IDPG: An Instance-Dependent Prompt Generation Method", NAACL 2022
>
> [3] Jiang et al., "Instance-wise Prompt Tuning for Pretrained Language Models", arXiv 2022

---

### Official Review · Reviewer_hgaH · 2023-08-07

**Soundness:** 3

**Excitement:**

3: Ambivalent: It has merits (e.g., it reports state-of-the-art results, the idea is nice), but there are key weaknesses (e.g., it describes incremental work), and it can significantly benefit from another round of revision. However, I won't object to accepting it if my co-reviewers champion it.

**Paper Topic And Main Contributions:**

Prompt tuning has emerged as a successful parameter-efficient alternative to the full fine-tuning of language models. However, prior works on prompt tuning often utilize long soft prompts of up to 100 tokens to improve performance, overlooking the inefficiency associated with extended inputs. This paper proposes a novel prompt tuning method SMoP (Sparse Mixture-of-Prompts) that utilizes short soft prompts for efficient training and inference while maintaining performance gains typically induced from longer soft prompts.

**Questions For The Authors:**

(1) Is there any reason why discrete prompts based approaches and adapter approaches are not used in the comparison?

(2) Regarding the number of tuned parameters, can the proposed approach use less parameters compared to adapter based approaches (while maintain competitive accuracy performance)?

(2) For examples, BERT, RoBERTa and GPT are used in the paper of the baseline P-tuning. However, T5 is used to report the baseline P-tuning’s results. If we use BERT, RoBERTa and GPT, can the proposed approach still outperform the baseline P-tuning?


**Reasons To Accept:**

(1) The paper studies an emerging and important topic.

(2) The developed solution is well motivated and has good intuitions.

(3) The paper is well written and easy to follow.

**Reasons To Reject:**

(1) The baselines in the paper seem to focus on the soft prompts. The paper may also include baselines based on discrete prompts (which show competitive performance) in the comparisons.

(2) The paper may also consider adapter-based baseline, such as LoRA, in the comparisons.

(3) The paper may be further improved, by considering other model backbones. This can validated the proposed method is more generic. For examples, BERT, RoBERTa and GPT are used in the paper of the baseline P-tuning.

**Reproducibility:**

4: Could mostly reproduce the results, but there may be some variation because of sample variance or minor variations in their interpretation of the protocol or method.

**Reviewer Confidence:**

3: Pretty sure, but there's a chance I missed something. Although I have a good feel for this area in general, I did not carefully check the paper's details, e.g., the math, experimental design, or novelty.

---

> ### Author Rebuttal · Authors · 2023-08-29
>
> Thank you for your encouraging words and insightful suggestions. Below are our responses, and we are keen to address any further concerns.
>
> **Reasons to reject 1, 2.  and Question 1. Comparison to discrete-prompts-based approaches and adapter approaches.**
>
> - In the context of searching for suitable prompts for a given task, prompt tuning (i.e., continuous prompt-based method) can be interpreted as an extension of discrete prompt-based methods that relax the constraints of discrete prompts [1]. As we have shown our improvements over conventional prompt tuning, we have excluded discrete prompt-based methods from our experiments.
> - While the adapter-based methods do represent a domain of research in parameter-efficient fine-tuning, we mainly considered prompt-tuning methods as our baselines to align with our primary motivation to alleviate the inefficiencies associated with the length of soft prompts during prompt tuning.
> - To address your concerns, we have conducted additional experiments with three adapter-based baselines, namely Adapter [2], AdapterFusion [3], and LoRA [4], and the results are shown below.
>
> | Method | Trainable % | BoolQ | CB| COPA| MultiRC | RTE | WiC | Avg. |
> |:---:|:---:|:---:|:---:|:---:|:---:|:---:|:---:|:---:|
> | Adapter (m=48) | 1.5800 | 81.0 | 95.5 | 60.3 | 80.3 | 75.8 | 67.0 | **76.6** |
> | AdapterFusion (m=48) | 0.7963 | 79.5 | 94.6 | 58.3 | 79.2 | 79.2 | 65.9 | 76.1 |
> | LoRA (rank=8) | 0.3954 | 79.0 | 90.5 | 59.7 | 80.0 | 77.9 | 66.9 | 75.7 |
> | Adapter (m=8) | 0.2806 | 79.0 | 90.2 | 58.0 | 80.0 | 73.6 | 65.1 | 74.3 |
> | **SMoP (l=5, k=4)** | **0.0083** | 79.4 | 94.6 | 58.3 | 79.6 | 77.5 | 65.2 | 75.8 |
>
> - In this table, "m" for Adapters represents the bottleneck dimension of the adapter layer, whereas "l and "k" for SMoP represent the utilized soft prompt length and the number of soft prompts. The methods are listed in descending order of trainable parameters.
>
> **Question 2. Regarding the number of tuned parameters, can the proposed approach use less parameters compared to adapter-based approaches (while maintain competitive accuracy performance)?**
> - Yes, as shown in the experimental results from our previous response, SMoP uses 47 to 190 times smaller number of trainable parameters compared to adapter-based approaches, while maintaining competitive performance. Additionally, when the difference in trainable parameter ratio narrows to a factor of 33x, SMoP outperforms Adapter on 5 tasks out of 6. This highlights the parameter efficiency of SMoP compared to adapter-based approaches.
>
> **Reason to reject 3 and Question 3. Generalizability of SMoP to other backbone models.**
>
> - We have followed Lester et al. (2021) [5] to consider T5 as the main backbone model, and thus have conducted experiments with P-tuning on T5 for comparison.
> - As you have suggested, to validate the generalizability of SMoP to other backbone models, we have conducted additional experiments on RoBERTa-large on 3 datasets as an example. The results are shown below.
>
> | Model | Method | CB | RTE | WiC | Avg. |
> |:---:|:---:|:---:|:---:|:---:|:---:|
> | RoBERTa -large | Full | 98.3 | 83.8 | 69.9 | 84.0 |
> |  | Prompt Tuning | 65.6 | 57.6 | 57.5 | 60.2 |
> |  | P-tuning | 69.6 | **60.3** | 61.0 | 63.6 |
> |  | SMoP | **72.4** | 59.7 | **66.1** | **66.1** |
>
> - In the experiments, we have swept over soft prompt length of {20, 100} for both Prompt Tuning and P-tuning, and used the setting of four prompts of length 5 for SMoP.
>  - These results demonstrate that SMoP outperforms P-tuning with a different backbone model, namely RoBERTa-large. We note that there is still a significant gap between the performance of full fine-tuning and SMoP, which we believe is a direction for future work.
> - Due to additional implementations and extensive hyperparameter search required for different backbone models, we have conducted a small portion of experiments as an example. We will proceed with experiments on other tasks and backbone models and include the results in the camera-ready version paper. Thank you for your precious suggestion.
>
> References
>
> [1] Liu et al., "Pre-train, Prompt, and Predict: A Systematic Survey of Prompting Methods in Natural Language Processing", ACM Computing Surveys 2023
>
> [2] Houlsby et al., “Parameter-Efficient Transfer Learning for NLP”, ICML 2019
>
> [3] Pfeiffer et al., “AdapterFusion: Non-Destructive Task Composition for Transfer Learning”. EACL 2021
>
> [4] Hu et al., “LoRA: Low-Rank Adaptation of Large Language Models”, ICLR 2022
>
> [5] Lester et al., "The Power of Scale for Parameter-Efficient Prompt Tuning", EMNLP 2021

---

### Meta-Review · Area_Chair_AhYH · 2023-09-15

**Recommendation:** 4

**Metareview:**

This work focuses on increasing the parameter efficiency of prompt tuning through the use of sparsely selected prompts. Authors managed to fit many interesting results in a short paper and provide convincing evidence during the rebuttal. All reviewers agree on the usefulness of the method and experimental evidence (which is almost at a long-paper level). I encourage authors to add the relevant work and comparison discussed during rebuttal (e.g. Adamix, Attempt).

---

### Decision · Program_Chairs · 2023-10-07

**Decision:**

Accept-Main

**Comment:**

This work focuses on increasing the parameter efficiency of prompt tuning through the use of sparsely selected prompts. Authors managed to fit many interesting results in a short paper and provide convincing evidence during the rebuttal. All reviewers agree on the usefulness of the method and experimental evidence (which is almost at a long-paper level). I encourage authors to add the relevant work and comparison discussed during rebuttal (e.g. Adamix, Attempt).